# SelectIT: Selective Instruction Tuning for LLMs via Uncertainty-Aware Self-Reflection

**Liangxin Liu**[1]   **Xuebo Liu**[1]*   **Derek F. Wong**[2]   **Dongfang Li**[1]
**Ziyi Wang**[1]   **Baotian Hu**[1]   **Min Zhang**[1]

[1]Institute of Computing and Intelligence, Harbin Institute of Technology, Shenzhen, China
[2]NLP[2]CT Lab, Department of Computer and Information Science, University of Macau
`lliangxin967@gmail.com`, {`liuxuebo,hubaotian,zhangmin2021`}`@hit.edu.cn`
`derekfw@um.edu.mo`, {`crazyofapple,ziyiwang676`}`@gmail.com`

## Abstract

Instruction tuning (IT) is crucial to tailoring large language models (LLMs) towards human-centric interactions. Recent advancements have shown that the careful selection of a small, high-quality subset of IT data can significantly enhance the performance of LLMs. Despite this, common approaches often rely on additional models or data, which increases costs and limits widespread adoption. In this work, we propose a novel approach, termed *SelectIT*, that capitalizes on the foundational capabilities of the LLM itself. Specifically, we exploit the intrinsic uncertainty present in LLMs to more effectively select high-quality IT data, without the need for extra resources. Furthermore, we introduce a curated IT dataset, the *Selective Alpaca*, created by applying SelectIT to the Alpaca-GPT4 dataset. Empirical results demonstrate that IT using Selective Alpaca leads to substantial model ability enhancement. The robustness of SelectIT has also been corroborated in various foundation models and domain-specific tasks. Our findings suggest that longer and more computationally intensive IT data may serve as superior sources of IT, offering valuable insights for future research in this area. Data, code, and scripts are freely available at `https://github.com/Blue-Raincoat/SelectIT`.

## 1   Introduction

Large language models (LLMs) have attracted much attention due to their impressive capabilities in following instructions and solving intricate problems (Touvron et al., 2023b,a; Achiam et al., 2023; Penedo et al., 2023). A crucial aspect of enhancing LLMs' performance is instruction tuning (IT), which involves the supervised adjustment of LLMs using pairs of instructional data, essential for refining the models' ability to accurately respond to human instructions. Recent groundbreaking research, such as the LIMA (Zhou et al., 2023), highlights the critical importance of instructional data quality over quantity. Contrary to the approach of merely increasing the dataset size, a carefully selected, smaller dataset of higher quality can significantly improve LLMs' performance.

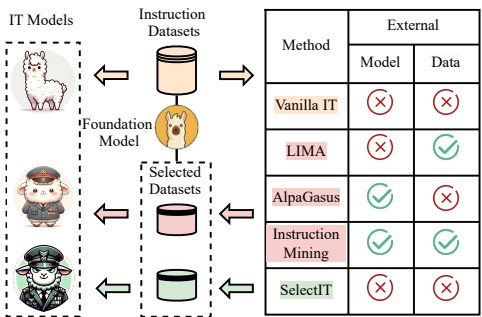

| Method | External | |
|---|---|---|
| | Model | Data |
| Vanilla IT | ⊗ | ⊗ |
| LIMA | ⊗ | ⊘ |
| AlpaGasus | ⊘ | ⊗ |
| Instruction Mining | ⊘ | ⊘ |
| SelectIT | ⊗ | ⊗ |

Figure 1: Existing advanced data selection strategies rely heavily on external models or data; however, SelectIT effectively overcomes this limitation.

---

* Corresponding Author

38th Conference on Neural Information Processing Systems (NeurIPS 2024).

Despite the development of various high-quality data selection methods, they often depend on external resources, limiting wider implementation. *External Model*: Chen et al. (2024); Liu et al. (2023) propose the employment of closed-source LLMs to evaluate or rank IT data. To circumvent the closed-source limitations, Li et al. (2023a,b); Kung et al. (2023) recommend fine-tuning open-source LLMs, which requires more computational resources. *External Data*: Cao et al. (2023) split all mixed data into several bins and fully trained the models to evaluate different indicators of high-quality IT data. Despite these advancements, the challenge of precise and efficient high-quality data selection without external resources remains unresolved.

In this paper, we introduce *SelectIT*, a novel approach designed to enhance IT data selection by fully leveraging the foundation model itself, eliminating the need for external resources. SelectIT employs different grain uncertainty of LLMs: token, sentence, and model, which can effectually improve the accuracy of IT data selection. We first use the foundation model itself to rate the IT data from 1 to $K$ based on the uncertainty of various tokens. Next, we use sentence-level uncertainty to improve the rating process by exploiting the effect of different prompts on LLMs. At a higher model level, we utilize the uncertainty between different LLMs, enabling a collaborative decision-making process for IT data selection. By applying SelectIT to the original Alpaca, we curate a compact and superior IT dataset, termed *Selective Alpaca*.

Experimental results show that SelectIT outperforms existing high-quality data selection methods, improving LLM's performance on the open-instruct benchmark (Wang et al., 2024). Further analysis reveals that SelectIT can effectively discard abnormal data and tends to select longer and more computationally intensive IT data. The primary contributions of SelectIT are as follows:

- We propose SelectIT, a novel IT data selection method which exploits the uncertainty of LLMs without using additional resources.
- We introduce a curated IT dataset, Selective Alpaca, by selecting the high-quality IT data from the Alpaca-GPT4 dataset.
- SelectIT can substantially improve the performance of LLMs across a variety of foundation models and domain-specific tasks.
- Our analysis suggests that longer and more computationally intensive IT data may be more effective, offering a new perspective on the characteristics of optimal IT data.

## 2 Related Work

**Instruction Tuning Dataset**    Recent empirical research highlights the substantial benefits of fine-tuning LLMs on specialized datasets containing instructions and responses, significantly enhancing their generalization capabilities and responsiveness to new questions (Chung et al., 2022; Longpre et al., 2023; Honovich et al., 2022; Sun et al., 2023). FLAN (Wei et al., 2022a) reformulates traditional natural language processing tasks as instructions formats, thereby improving model performance. Alpaca (Taori et al., 2023; Peng et al., 2023a) exemplifies the effectiveness of merging a select set of manual instruction seeds with advanced LLMs, like text-davinci-003 or GPT-4, to compile a comprehensive dataset. Similarly, Vicuna (Chiang et al., 2023) leverages 70,000 conversations from ChatGPT interactions, benefiting from the diverse data types and structures within these dialogues. WizardLM (Xu et al., 2023) introduces a novel approach by using LLMs to automatically generate open-domain instructions of varying complexities, achieving controlled instructional difficulty variation. However, LIMA (Zhou et al., 2023) demonstrates that only $1K$ high-quality IT data can match or exceed the performance of LLMs fine-tuned on larger IT datasets, presenting a promising direction for future research.

**Instruction Data Selection**    The recognition of IT data quality's superiority over quantity in the context of IT is well-established, yet the efficient and precise identification of high-quality data continues to be a challenging frontier for research. One straightforward approach is utilizing the closed-source advanced LLMs for IT data evaluation and selection (Chen et al., 2024; Liu et al., 2023). To circumvent the constraints associated with closed-source, existing research opt to fine-tune LLMs directly to select high-quality IT data (Li et al., 2023b; Kung et al., 2023). Li et al. (2023c); Gururangan et al. (2020); Chen et al. (2023a); Cao et al. (2023) use pre-defined notions of useful data or other IT datasets to develop a data quality assessment framework. Li et al. (2023a) propose training a specialized model and utilizing two unique, condition-based losses on this for a comprehensive IT

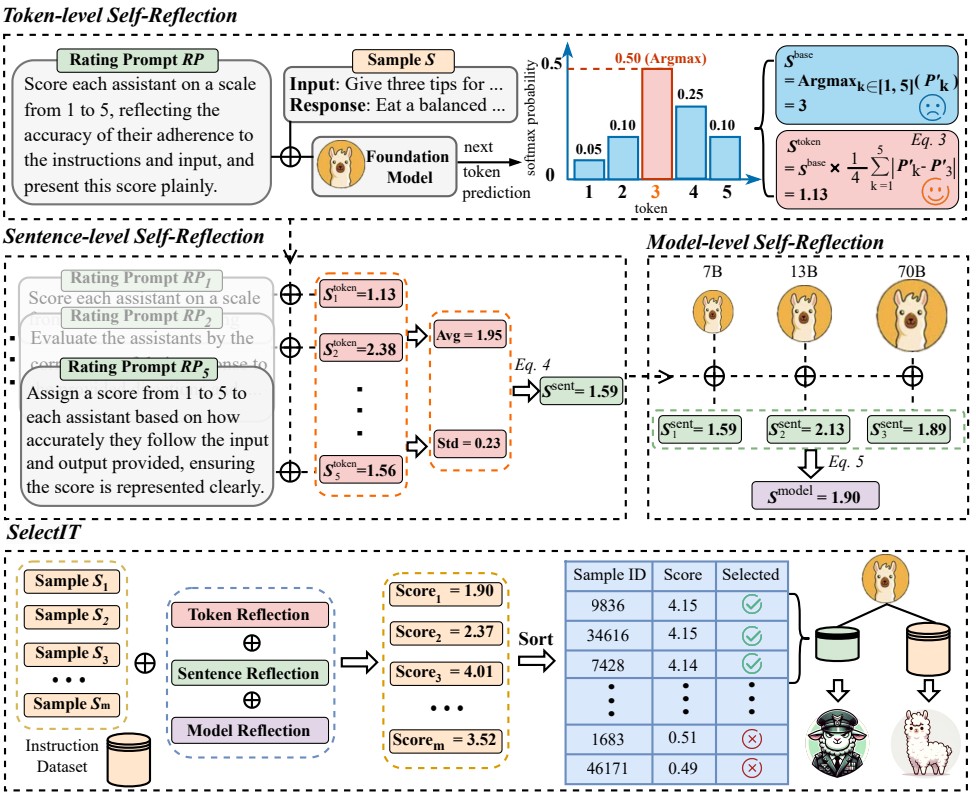

Figure 2: Overall framework of SelectIT. In Token-level Self-Reflection, we employ the foundation model to rate the IT data from 1 to $K$. In Sentence-level Self-Reflection, we leverage the uncertainty of varied prompts on LLMs to enhance the rating process. In Model-level Self-Reflection, we harness uncertainty among different LLMs to facilitate a collaborative decision-making process in selecting IT data. Finally, different levels of self-reflection are reasonably combined into SelectIT, which can effectively select high-quality IT data without relying on additional resources.

data selection. Wu et al. (2023) explore where data selection is informed by the similarity of samples within the embedding space of a fine-tuned model. N-gram features (Xie et al., 2023) or model gradients (Xia et al., 2024; Han et al., 2023) are also important features for selecting high-quality data in fine-tuned LLMs. However, the methods described above depend, to varying degrees, on supplementary datasets, the use of closed-source models, or open-source models that have been specially fine-tuned, which results in increased consumption of resources and potentially limits the broader impact.

## 3 Our SelectIT Method

Utilizing advanced LLMs for the sample evaluation is a widely adopted approach in the IT data selection (Chen et al., 2024; Li et al., 2023b; Liu et al., 2023). Given an IT dataset $D$ containing a sample $S = $ (input $X$, response $Y$), a designated rating prompt $RP$, and the foundation LLMs $M$, the goal is to leverage both $RP$ and $S$ to prompt $M$ to assign an evaluation $Score$ to the sample $S$ on a scale from 1 to $K$. A higher score typically signifies superior IT data $Quality$.

$$Quality \propto Score \in [1, K] = M(RP, S) \tag{1}$$

While existing methods (Chen et al., 2024; Cao et al., 2023) are adept at identifying high-quality samples, they often over-rely on external resources. To address these challenges, we introduce SelectIT, a strategy that capitalizes on the internal uncertainty of LLMs to efficiently select high-quality IT data. SelectIT incorporates three grains of sample evaluation modules: token, sentence, and model-level self-reflections, which effectively improve the reliability of IT data selection. The comprehensive framework of SelectIT is depicted in Figure 2.

### 3.1 Token-level Self-Reflection

Numerous studies have demonstrated that foundation models exhibit robust capabilities for next-token prediction during their pre-training phase (Touvron et al., 2023b,a). Yet, this predictive strength is frequently underutilized in evaluating IT data quality. In SelectIT, we adopt a similar idea to evaluate IT data. Specifically, we calculate the next-token probability (from 1 to $K$) based on the rating prompt $RP$ and sample $S$. The score token with the highest probability is then considered as the sample's quality.

$$S^{base} = \underset{k \in \{1,...,K\}}{\arg\max} P'_k, P'_k = \left( \frac{P_k}{\sum_{j=1}^{K} P_j} \right) \tag{2}$$

where $P_k$ and $P'_k$ mean the probability and normalized probability of token $k$.

The probability distribution among score tokens reflects the internal uncertainty of LLMs on sample evaluation. The higher $P'_{S^{base}}$, the more confidence of LLMs, which is not well exploited in Equation 2. To capture this subtle difference, we introduce the token-level self-reflection (Token-R), which uses the distribution between tokens that reflect the internal uncertainty of LLMs, to enhance the credibility of quality assessment. Specifically, we assess the average disparity between the predicted $S^{base}$ token and the other, where the greater the disparity, the more the confidence of LLMs. This disparity is then utilized to refine the original $S^{base}$, resulting in a token-level score $S^{token}$.

$$S^{token} = S^{base} \times \underbrace{\frac{1}{K-1} \sum_{i=1}^{K} |P'_i - P'_{S^{base}}|}_{Uncertainty} \tag{3}$$

### 3.2 Sentence-level Self-Reflection

Different prompts can significantly affect outputs of LLMs (Kung et al., 2023; Peng et al., 2023b), introducing uncertainty into IT data evaluation at the sentence level. To make better use of this uncertainty to bolster the reliability of our method, we implement sentence-level self-reflection (Sentence-R). Building upon Token-R, we devise $K$ semantically similar rating prompts $\{RP_0, RP_1, \ldots, RP_K\}$ to obtain a series of quality scores $\{S_0^{token}, S_1^{token}, \ldots, S_K^{token}\}$ based on a given sample $S$. We calculate the average of these scores to represent the overall quality of sample $S$, because of the importance of incorporating assessments from diverse prompts. Additionally, we use the standard deviation to quantify the LLMs' uncertainty to rating prompt; a higher standard deviation suggests greater sensitivity to prompt variation, while a lower standard deviation indicates more consistent and confident quality ratings by LLMs (Zhou et al., 2020). By integrating a holistic sample evaluation with the quantification of model uncertainty, we derive the sentence-level score $S^{sent}$, offering a more nuanced and reliable measure of IT data quality.

$$S^{sent} = \frac{\mathbf{Avg}\{S_i^{token}\}_{i=1}^{K}}{1 + \alpha \times \underbrace{\mathbf{Std}\{S_i^{token}\}_{i=1}^{K}}_{Uncertainty}} \tag{4}$$

where $\mathbf{Avg}\{\cdot\}$ and $\mathbf{Std}\{\cdot\}$ respectively denote the mean and standard deviation of $S_i^{token}$, K means the number of rating prompts $RP$. Moreover, we use the uncertainty factor $\alpha$ to control for the impact of the uncertainty of LLMs on overall scores.

### 3.3 Model-level Self-Reflection

A sample affirmed by multiple foundation models can truly be deemed as high-quality. Different foundation models have different quality assessments of the sample, which introduce model-level uncertainty. To maximize the utilization of this uncertainty, we introduce model-level self-reflection (Model-R). This strategy leverages the capabilities of existing open-source models without the need for additional resources or the complexities associated with fine-tuning. However, the challenge lies in the diverse capabilities of various LLMs and determining how to reasonably combine their sample evaluation based on their performance. It is widely acknowledged that the capabilities of LLMs tend to increase with their parameter count (Hendrycks et al., 2021). Thus, we suggest using the parameter

count of LLMs as an initial metric for assessing their capabilities to properly weight sample quality scores. Given $N$ foundation models with parameter counts $\{\theta_1, \theta_2, \ldots, \theta_N\}$ and their respective sentence-level scores for a sample $S$ being $\{S_0^{sent}, S_1^{sent}, \ldots, S_N^{sent}\}$, we formulate the model-level score $S^{model}$ to reflect a comprehensive evaluation of sample quality.

$$Quality \propto S^{model} = \sum_{i=1}^{N} \left( \frac{\theta_i}{\sum_{j=1}^{N} \theta_j} \times S_i^{sent} \right) \tag{5}$$

where $N$ means the number of the foundation models. By obtaining LLM parameters without resource expenditure, Model-R effectively allows us to employ more powerful foundation models, which is advantageous for selecting higher-quality data. Finally, we use $S^{model}$ as the final evaluation of sample $S$ in SelectIT. The higher $S^{model}$, the better sample quality. We sort the samples in descending order based on their $S^{model}$ and then select the top-ranked samples as high-quality data.

### 3.4 Selective Alpaca

We apply SelectIT to the widely-used Alpaca-GPT4 (Peng et al., 2023a). Specifically, we use the most popular LLaMA-2 (7B, 13B, 70B) as our foundation models and set the hyper-parameters $\alpha = 0.2$ and $K = 5$, which decides the range of LLMs rating in Token-R and the number of rating prompts in Sentence-R. We finally select the top 20%, a total of 10.4K pairs as the high-quality data and obtain a curated IT dataset called *Selective Alpaca*.

## 4 Experiments

### 4.1 Setups

**Benchmark**  To gain a more comprehensive understanding of the capabilities of LLMs, we evaluate our approach in diverse downstream tasks (Wang et al., 2024; Ivison et al., 2023). *Factual knowledge*: We use the Massive Multitask Language Understanding dataset (MMLU (Hendrycks et al., 2021)) to assess the factual knowledge of LLMs and report 5-shot results. *Reasoning*: We evaluate the reasoning abilities of LLMs using two widely utilized datasets: the Grade School Math dataset (GSM (Cobbe et al., 2021)) and Big-Bench-Hard (BBH (Suzgun et al., 2022)) with the CoT setting (Wei et al., 2022b). *Multilinguality*: we assess this ability by TyDiQA, a multilingual question-answering benchmark that encompasses 11 diverse languages, with the gold-passage setup. *Coding*: We evaluate this ability using the HumanEval dataset (Chen et al., 2021) and report pass@10 results with a temperature of 0.8. *Open-ended generation*: We utilize AlpacaEval (Dubois et al., 2023), which employs GPT-4 to effectively assess model outputs. This can evaluate whether the text produced by LLMs aligns with humans.

**Implementation Details**  We use LLaMA-2 as our testbed. We fine-tune it for 3 epochs, with a batch size of 128. We use Adam with $\beta_1 = 0.9$, $\beta_2 = 0.999$, and the cosine learning rate scheduler starts from $2e-5$, and decays to 0. we opted for a 4096 input length because it can show the best performance of LLMs. We employ the beam $= 4$ for decoding. We set the temperature parameter to 0.8 and the top$-$p sampling parameter to 0.9 to improve the originality of the output text while ensuring the accuracy and relevance of the content.

**Baselines**  We compare with the following baselines:

- **Alpaca-GPT4** (Peng et al., 2023a) is a widely-used IT dataset that implements a self-instruct method to autonomously generate instructions by the advanced GPT4.
- **LIMA** (Zhou et al., 2023) primarily consists of 1000 manually crafted high-quality instructional data, which can better stimulate the alignment capability of LLMs.
- **AlpaGasus** (Chen et al., 2024) involves utilizing the robust ChatGPT to score and select data from the original Alpaca-GPT4 dataset.
- **Q2Q** (Li et al., 2023a) operates by training a precursor model, determining the quality of the IT data based on the two different loss values within this model.
- **Instruction Mining** (Cao et al., 2023) entails fitting data features and loss values to derive a formula for assessing data quality.

| ID | System | External Model | External Data | MMLU | BBH | GSM | TydiQA | CodeX | AE | Overall AVG | Overall Δ (↑) |
|---|---|---|---|---|---|---|---|---|---|---|---|
| | *Base Model: LLaMA-2-7B* | | | *Implemented Existing Method* | | | | | | | |
| 1 | Alpaca-GPT4 | | | 46.5 | 38.4 | 15.0 | 43.4 | 26.8 | 34.2 | 34.1 | - |
| 2 | LIMA | ✗ | ✓ | 45.4 | 37.5 | 14.3 | 45.1 | 24.6 | 33.1 | 33.3 | -0.7 |
| 3 | 1 + AlpaGasus | ✓ | ✗ | 45.9 | 39.0 | 14.5 | 46.4 | 27.5 | 35.4 | 34.8 | +0.7 |
| 4 | 1 + Q2Q | ✓ | ✗ | 46.9 | 39.4 | 15.3 | 46.7 | 28.2 | 35.7 | 35.4 | +1.3 |
| 5 | 1 + Instruction Mining | ✓ | ✓ | 47.0 | 39.6 | 16.5 | 47.1 | 28.6 | 34.4 | 35.5 | +1.5 |
| | | | | *Our Proposed Method (Individual)* | | | | | | | |
| 6 | 1 + Token-R | ✗ | ✗ | 46.8 | 36.5 | 14.5 | 44.6 | 28.9 | 35.5 | 34.5 | +0.4 |
| 7 | 1 + Sentence-R | ✗ | ✗ | 46.9 | 38.1 | 16.1 | **48.4** | 26.9 | 35.3 | 35.3 | +1.2 |
| 8 | 1 + Model-R | ✗ | ✗ | 47.3 | 37.4 | 16.1 | 45.3 | 28.4 | **35.8** | 35.1 | +1.0 |
| | | | | *Our Proposed Method (All)* | | | | | | | |
| 9 | SelectIT (6 + 7 + 8) | ✗ | ✗ | **47.4** | **40.6** | **16.8** | 47.4 | **29.4** | 35.7 | **36.2** | **+2.2** |
| | *Base Model: LLaMA-2-13B* | | | *Implemented Existing Method* | | | | | | | |
| 10 | Alpaca-GPT4 | | | **55.7** | 46.6 | 30.5 | 48.1 | 40.8 | 46.5 | 44.7 | - |
| 11 | LIMA | ✗ | ✓ | 54.6 | 45.3 | 30.5 | 51.1 | 34.1 | 42.6 | 43.0 | -1.7 |
| 12 | 10 + AlpaGasus | ✓ | ✗ | 54.1 | 47.3 | 31.5 | 50.6 | 41.3 | 46.3 | 45.2 | +0.5 |
| 13 | 10 + Q2Q | ✓ | ✗ | 55.3 | 48.5 | 32.0 | 50.8 | 41.3 | 47.3 | 45.9 | +1.2 |
| 14 | 10 + Instruction Mining | ✓ | ✓ | 54.1 | 47.3 | 32.5 | 52.6 | **43.3** | 48.3 | 46.3 | +1.6 |
| | | | | *Our Proposed Method (Individual)* | | | | | | | |
| 15 | 10 + Token-R | ✗ | ✗ | 55.3 | 47.3 | 30.5 | 51.3 | 39.8 | 46.2 | 45.1 | +0.4 |
| 16 | 10 + Sentence-R | ✗ | ✗ | 55.2 | 48.3 | 31.0 | 52.2 | 42.5 | 46.3 | 45.9 | +1.2 |
| 17 | 10 + Model-R | ✗ | ✗ | 55.1 | 47.5 | 31.5 | 52.3 | 40.2 | 46.1 | 45.5 | +0.8 |
| | | | | *Our Proposed Method (All)* | | | | | | | |
| 18 | SelectIT (15 + 16 + 17) | ✗ | ✗ | 55.7 | **48.9** | **33.0** | **54.1** | 42.2 | **48.8** | **47.1** | **+2.4** |

Table 1: Overall results on IT. "CodeX" and "AE" mean HumanEval and AlpacaEval benchmarks. All the scores are averages of three independent runs with different random seeds.

## 4.2 Main Results

We focus on the discussion of LLaMA-2-13B because both 7B and 13B models exhibit similar trends in Table 1. System (10) shows the vanilla IT on LLMs with the original Alpaca. By using the data selection strategies, the ability of LLMs has a moderate enhancement in Systems (12) to (14). Additionally, we can use $S^{base}$ as the input for Equations 4 and 5 to construct individual methods of Sentence-R and Model-R. Systems (15) to (17) illustrate that applying each submodule of SelectIT incrementally enhances LLMs' performance, rivaling contemporary advanced methods.

Most remarkably, SelectIT can better boost LLaMA-2's performance compared to vanilla IT in the System (18). Compared to other IT data selection strategies, this enhancement is particularly evident in the computational and reasoning tasks on the BBH and GSM benchmarks. This may be attributed to the characteristics of selected data by SelectIT, and we will analyze this phenomenon in a later section. These gains in reasoning ability also positively impact the coding proficiency of LLMs. The improvement of LLMs on the TydiQA dataset is also obvious enough, which shows that SelectIT can effectively eliminate similar samples and retain sufficient diversity in multilingual aspects.

## 5 Analysis

This part aims to answer the research questions through the following experiments: How to select high-quality data in SelectIT? (§5.1) Is SelectIT adaptable to various models and domains? (§5.2) How about the efficiency of SelectIT? (§5.3) What are the advantages of Selective Alpaca?(§5.4)

### 5.1 Abalation Study of SelectIT

**Effect of IT Data Quantity** While SelectIT already excels at assessing and ranking samples effectively, selecting an appropriate number of samples in a redundant dataset remains a crucial aspect of our method. We divide the Alpaca dataset into multiple subsets ranging from 10% to 100% based on SelectIT's evaluation and

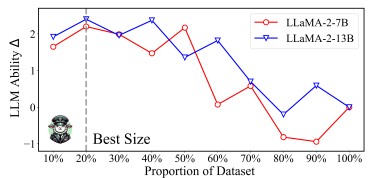

Figure 3: Comparison of LLM abilities with varying Alpaca proportions.

evaluate the overall ability of LLMs on the open-instruct benchmark. As illustrated in Figure 3, compared to using the full Alpaca dataset, we observe that LLMs achieve optimal performance using the top 20% to 40% data. Hence, considering the tradeoff of training resources, training time, and model performance, we opt for 20% for implementing the SelectIT on the Alpaca dataset.

**Effect of Multiple Rating Prompts** $K$ is a critical parameter for our method, impacting not only the range of scores assigned by the LLMs but also the number of rating prompts. We set $K = 3, 5, 7, 9$ and apply SelectIT for sample selection within the Alpaca to get different subset datasets. Table 2 indicates that variations in the value of $K$ have a minor impact on the overall performance of the LLMs. This is attributed to

| $K$ | LLaMA-2-7B | LLaMA-2-13B | Overall |
|---|---|---|---|
| 3 | 35.6 | 46.4 | 40.5 |
| 5 | **36.2** | 47.1 | **41.7** |
| 7 | 35.7 | **47.3** | 41.5 |
| 9 | 36.0 | 46.8 | 41.4 |

Table 2: Effect of different $K$.

our multi-granularity self-reflection mechanism, which effectively enhances the accuracy and stability of sample selection. Although the model achieves competitive performance at $K$ values of 5 and 7, to minimize resource consumption, we set $K = 5$ as the default value in SelectIT.

**Effect of Uncertainty** $\alpha$ is an uncertainty factor, integral to calibrating the equilibrium between the mean and the standard deviation of scores derived from Token-R. We assign $\alpha$ four different values, i.e., 0.2, 0.4, 0.6, and 0.8, and incorporate SelectIT for sample selection from within Alpaca to generate disparate subset datasets, with all other parameters remaining constant. As shown in Table 3, with a rise in the

| $\alpha$ | MMLU | BBH | GSM | Tydiqa | CodeX | AE | AVG |
|---|---|---|---|---|---|---|---|
| 0.2 | 47.4 | **40.6** | 16.8 | **47.4** | 29.4 | 35.7 | **36.2** |
| 0.4 | **47.9** | 39.4 | 15.5 | 46.5 | 29.4 | **35.8** | 35.8 |
| 0.6 | 47.8 | 39.8 | 16.5 | 45.6 | 29.1 | 35.1 | 35.7 |
| 0.8 | 47.6 | 36.4 | 16.5 | 43.6 | 26.7 | 35.4 | 34.4 |

Table 3: Effect of different $\alpha$.

$\alpha$ value, Sentece-R tends to emphasize the uncertainty innate to LLMs. This results in the neglect of the average score, a fundamental indicator of sample quality, thereby contributing to a decrease in the overall performance of LLMs. Consequently, we ascertain that an $\alpha$ value of 0.2 is optimally suited to establish an effective balance between the sample's quality and the model's uncertainty.

**Effect of Different Reflection Strategy** We analyze the relationship between individual selection strategies and SelectIT, from the following two aspects. We first account for the number of high-quality data that can only be selected by a unique selection strategy, referred to as unique selection. Secondly, we calculate which samples in Selective Alpaca can be selected by individual selection strategies in Selective Alpaca, called overall selection. As shown in Table 4, Sentence-R plays the most important role in the final SelectIT strategy. This is because rating prompts play an important role in sample evaluation and exploiting the effect of

| ID | Individual | Unique (%) | Overall (%) |
|---|---|---|---|
| 6 | Token-R | 6.18 | 17.83 |
| 7 | Sentence-R | **40.81** | **63.98** |
| 8 | Model-R | 7.37 | 23.08 |

Table 4: The relationship between the SelectIT and the individual selection strategy. Sentence-R plays the most significant impact on the final rating of the IT data. IDs 6, 7, and 8 correspond to the system of the same IDs in Table 1.

different prompts on LLM can effectively better improve the accuracy of sample evaluation than Token-R and Model-R. Additionally, this phenomenon also aligns with the model's performance reported in Table 1, showing the rationality of our proposed uncertainty-aware self-reflection methods.

| Base Model | Datasets | Data Size | MMLU | BBH | GSM | Tydiqa | CodeX | AE | Overall | |
|---|---|---|---|---|---|---|---|---|---|---|
| | | | | | | | | | AVG | $\Delta$ ($\uparrow$) |
| LLaMA-2-7B | LIMA | 1K | 45.4 | 37.5 | 14.3 | 45.1 | 24.6 | 33.1 | 33.3 | - |
| | Selective Alpaca | 1K | **46.6** | **41.3** | **14.5** | **46.2** | **30.6** | **33.8** | **35.5** | +2.2 |
| | AlpaGasus | 9K | 45.9 | 39.0 | 14.5 | **46.4** | 27.5 | 35.4 | 34.8 | - |
| | Selective Alpaca | 9K | **47.2** | **41.3** | **18.5** | 47.6 | **28.3** | 35.4 | **36.4** | +1.6 |

Table 5: Results on IT for different datasets with the same number of instances.

**Effect of Data Imbalance** To eliminate unfair comparison caused by IT data quantity imbalance, we adjust the size of the Selective Alpaca dataset to 1,000 and 9,229 respectively, aligning with the LIMA (Zhou et al., 2023) and AlpaGasus (Chen et al., 2024) datasets. The results in Table 5 show

| Base Model | Datasets | MMLU | BBH | GSM | Tydiqa | CodeX | AE | Overall | |
|---|---|---|---|---|---|---|---|---|---|
| | | | | | | | | AVG | Δ (↑) |
| **LLaMA-2-7B** | Alpaca-GPT4 | 46.5 | 38.4 | 15.0 | 43.4 | 26.8 | 34.2 | 34.1 | - |
| | Selective Alpaca | **47.4** | **40.6** | **16.8** | **47.4** | **29.4** | **35.7** | **36.2** | **+2.1** |
| **LLaMA-2-13B** | Alpaca-GPT4 | **55.7** | 46.6 | 30.5 | 47.1 | 38.8 | 46.5 | 44.2 | - |
| | Selective Alpaca | 55.3 | **48.5** | **32.5** | **54.1** | **41.2** | **47.8** | **46.6** | **+2.4** |
| **Mistral-7B** | Alpaca-GPT4 | 52.5 | 51.7 | 33.5 | **51.1** | 54.7 | 43.1 | 47.8 | - |
| | Selective Alpaca | **56.9** | **53.7** | **36.0** | 49.3 | **55.3** | **44.3** | **49.3** | **+1.5** |
| **LLaMA-3-8B** | Alpaca-GPT4 | 59.6 | 52.3 | 34.5 | **43.1** | 60.2 | **48.2** | 49.7 | - |
| | Selective Alpaca | **61.2** | **55.0** | **37.5** | 41.1 | **65.4** | 47.7 | **51.3** | **+1.6** |

Table 6: Results of IT with various foundation models.

| Datasets | Data Size | MMLU | BBH | GSM | Tydiqa | CodeX | AE | Overall | |
|---|---|---|---|---|---|---|---|---|---|
| | | | | | | | | AVG | Δ (↑) |
| WizardLM | 143K | 43.8 | 37.8 | 10.0 | 41.2 | 25.2 | **35.3** | 32.2 | - |
| WizardLM + SelectIT | 28.6K | **45.1** | **40.1** | **11.0** | **43.1** | **27.5** | 34.7 | **33.6** | **+1.4** |
| Orca-GPT4 | 1M | 40.1 | 35.6 | 13.0 | **46.0** | 23.3 | **38.1** | 32.7 | - |
| Orca-GPT4 + SelectIT | 0.2M | **43.9** | **38.7** | **16.5** | 42.0 | **27.7** | 37.4 | **34.4** | **+1.7** |

Table 7: Results of IT with various IT datasets.

that, when facing the same amount of data, SelectIT can still demonstrate better performances, which further illustrates its effectiveness.

## 5.2 Robustness across Models, Datasets and Domains

**Various Foundation Models**   Although Selective Alpaca achieved impressive improvements in LLaMA-2, applying it to other foundation models remains a challenging task. To address this, we apply Selective Alpaca on the Mistral-7B and LLaMA-3-8B LLMs and present our results on the open-instruct benchmark alignment with the above test configuration. As depicted in Table 6, although Selective Alpaca is selected by the LLaMA-2 models, it is also applicable to the Mistral-7B, LLaMA-3-8B and improves their capabilities across various tasks, especially on MMLU, BBH, and GSM benchmarks. This experiment fully demonstrates the flexibility of SelectIT which does not rely on a specific foundation model for data selection and the universality of Selective Alpaca which can effectively improve the capabilities of different series or scale LLMs.

**Various Instruction Tuning Datasets**   We further validate the robustness of SelectIT by deploying it on two additional, widely-utilized datasets: WizardLM (Xu et al., 2023) and Orca-GPT4 (Subhabrata & Arindam, 2023). WizardLM introduces an innovative method of using LLMs to auto-generate open-domain instructions of varying complexities. This allows for a controlled variation in instructional difficulty and the dataset comprises 143K samples. Orca-GPT4 on the other hand, leverages rich signals from GPT-4 that include explanation traces, step-by-step thought processes, and other multifaceted instructions, all under the guidance of teacher assistance from ChatGPT. Additionally, we maintain consistent hyperparameters, such as $\alpha$ and $K$, choosing LLaMA-2-7B as our base model. We limit the fine-tuning of these datasets to one epoch. As shown in Figure 7, SelectIT consistently enhances the performance of the model on both the WizardLM and Orca-GPT4 datasets. Notably, this augmentative effect is especially pronounced in the computational and reasoning tasks within the BBH and GSM benchmarks. In evaluating three separate IT datasets, specifically Alpaca-GPT4, WizardLM, and the more extensive Orca-GPT4, our extensive experimental conclusions validate the broad utility and durability of SelectIT.

**Various Domain-specific Tasks**   Machine translation (MT) is a representative domain-specific task of LLMs. Previous works have already demonstrated significant improvements with LLMs, but they usually use redundant translation IT datasets. This part tests the robustness of SelectIT on the IT dataset of MT. We select the powerful MT LLM ALMA (Xu et al., 2024) as our backbone model.

We choose the representative language pairs {German, Chinese}⇔English from WMT'17 to WMT'20 human-written test datasets, and development and test sets from Flores-200, totaling 30K training examples. We used WMT'22 test data for testing, and finally, 6K high-quality

examples were selected using SelectIT. We utilize both BLEU (Post, 2018; Ott et al., 2018) and COMET (Rei et al., 2022) based on the *wmt22-comet-da* model for evaluation. We report results for the two language pairs in four directions, using ALL to represent their average. Table 8 shows that SelectIT consistently improves ALMA's translation performance. These results indicate that SelectIT is a versatile and scalable method, effective not only for IT data selection but also for domain-specific tasks like MT. For more detailed analysis and results, please see Appendix A.1.

| Method | Size | ALL | |
|---|---|---|---|
| | | COMET | BLEU |
| *SoTA Models* | | | |
| NLLB (Costa-jussà et al., 2022) | 54B | 78.8 | 26.3 |
| GPT-3.5 | - | 85.6 | 34.8 |
| GPT-4 | - | 85.8 | 35.1 |
| *Existing Method* | | | |
| LLaMA-2 (Touvron et al., 2023b) | 7B | 76.5 | 21.1 |
| TIM (Zeng et al., 2023) | 7B | 79.1 | 26.4 |
| SWIE (Chen et al., 2023b) | 7B | 80.6 | 27.6 |
| BigTranslate (Yang et al., 2023) | 13B | 78.8 | 21.9 |
| Bayling (Zhang et al., 2023) | 13B | 82.0 | 27.8 |
| *Our Implemented Method* | | | |
| ALMA | 7B | 83.2 | 29.7 |
| w/ SelectIT | 7B | **83.7** | **30.5** |
| ALMA | 13B | 83.7 | 31.5 |
| w/ SelectIT | 13B | **84.2** | **32.2** |

Table 8: The overall results on MT LLMs.

## 5.3 Efficiency of SelectIT

SelectIT is a faster and more cost-effective method for IT data selection. We compared different selection methods on the Alpaca-GPT4 dataset. For ChatGPT (AlpaGasus) or GPT-4, we randomly select 500 instruction data from Alpaca-GPT4, analyze various metrics, and estimate the resource consumption for selecting the entire dataset. Using SelectIT, we employ 4 A800 80G

| Method | Speed | Time | Cost |
|---|---|---|---|
| ChatGPT API | 0.76 it/s | 19.07h | $52.02 |
| GPT4 API | 0.37 it/s | 38.98h | $2871.56 |
| SelectIT | **9.34 it/s** | **5.80h** | **$26.68** |

Table 9: Comparison of selection efficiency.

GPUs to select high-quality IT data, calculating the total cost based on Google Cloud's rate of $1.15/h per single GPU. As shown in Table 9, SelectIT is significantly faster and uses the least resources. This efficiency is due to computing only the probability of the next token for input sentences, bypassing the full sentence generation and decoding process, resulting in lower resource consumption. Additionally, using our own GPU at a low cost enhances transparency, allowing us to preserve all intermediate outputs and results for thorough analysis in data selection.

## 5.4 Insights of Selective Data Curation

**Different Selection Strategies** This part compares three different selection strategies, namely, randomly selecting 20% in the full Alpaca and unselected dataset of Selective Alpaca, and selecting 20% data based on sample length (Zhao et al., 2024). As shown in Table 10, the random-based strategies show certain performance degradation and the random selection in the unselected dataset is even worse, which reflects the effectiveness

| Method | LLaMA-2 | | ALMA | | Δ (↑) |
|---|---|---|---|---|---|
| | 7B | 13B | 7B | 13B | |
| Full Dataset | 34.1 | 44.2 | 29.7 | 31.5 | - |
| w/ Random (Full) | 34.1 | 45.1 | 29.3 | 31.0 | 0.0 |
| w/ Random (Unselected) | 34.6 | 44.3 | 29.1 | 31.2 | -0.4 |
| w/ Length | 35.5 | 47.1 | 30.1 | 31.8 | +5.0 |
| w/ SelectIT | **36.2** | **47.1** | **30.5** | **32.2** | **+6.5** |

Table 10: Comparasion with variants.

of our method from the side. Selection based on sample length is a simple approach to defining high-quality data, but it does not take into account the content of IT data, resulting in the limited performance of LLMs. SelectIT can significantly improve the abilities of LLMs.

**Data Representation Analysis** This part explores the relationship between Selective Alpaca and the original datasets from a representation perspective. Following Gao et al. (2024), we use the outputs of the last layer corresponding to the last token in the input sequence as sample representations. We then apply T-SNE (Hinton & Roweis, 2002) for dimensionality reduction, mapping high-dimensional embeddings onto a 2D space. Figure 4 shows the intermediate representations generated by the full and Selective Alpaca datasets. Randomly selected data struggle to distinguish abnormal data far from the center, making it hard to define high-quality IT data. In contrast, Selective Alpaca data are mostly concentrated around the center, indicating that our dataset predominantly contains high-quality data near the center and effectively discards abnormal data, supporting the conclusion of Table 10.

**Data Characteristic Analysis** We analyze the Selective Alpaca from the following two perspectives, to explore why our dataset is better than the original dataset and its variants. Firstly, as shown in Figure 5, the length of instructions from the Selective Alpaca is significantly longer than those in the Alpaca dataset and AlpaGasus which is selected by ChatGPT. This implies that, with the same amount of data, our dataset contains more information, aligned with the results in Table 10. Secondly, by using ChatGPT to examine IT data types, we find a substantial increase in the proportion of computational problems in Selective Alpaca. This indicates that Selective Alpaca tends to select

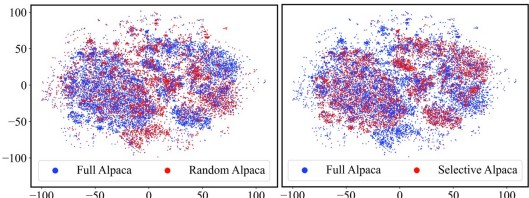
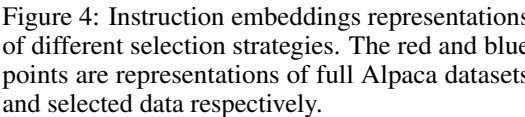

Figure 4: Instruction embeddings representations of different selection strategies. The red and blue points are representations of full Alpaca datasets and selected data respectively.

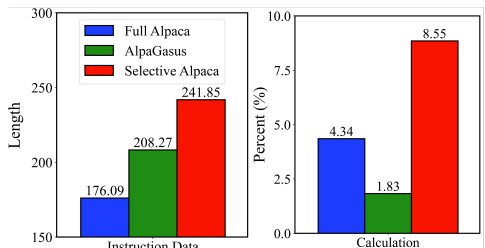

Figure 5: Left: The average length of samples. Right: The proportion of calculation type.

high-quality mathematical data, providing a solid explanation for the observed improvement in the reasoning abilities of LLMs as demonstrated in Table 1. Appendix A.3 shows the case study of comparing the Selective Alpaca with AlpaGasus.

**Insights of High-Quality Data in SelectIT** Furthermore, we analyze the proportion of calculation and sample average length in Alpaca-GPT4 with different proportions after sorting by SelectIT to explore its intrinsic characteristics and the definition of high-quality data. As shown in Figure 6, with the proportion of Alpaca-GPT4 data continuing to increase, the proportion of calculation and sample average length gradually decreases. This phenomenon clearly indicates that SelectIT can reasonably rank samples based on their characteristics. When the data size is more than 50%, the proportion of calculation IT data sharply declines, falling below 6%, causing a noticeable decrease in the model's overall capability, as depicted in Figure 3. This analysis shows that more computationally intensive IT data may be a new perspective on the characteristics of optimal IT data, which not only effectively improves the LLMs' reasoning ability, but also further drives the improvement of other abilities.

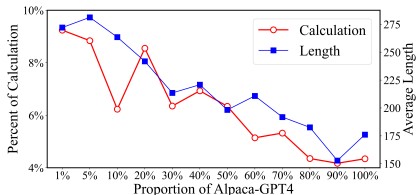

Figure 6: Changing trends of the calculation and sample length with different data sizes.

# 6 Conclusion

This paper introduces a novel data selection strategy, SelectIT, for LLM instruction tuning, which uses LLM uncertainty to efficiently identify high-quality IT data without requiring additional resources. SelectIT includes three types of self-reflection: token, sentence, and model, which can individually and jointly improve the performance of IT data selection. By applying SelectIT to the Alpaca-GPT4 dataset, we introduce a compact and strong IT dataset, called Selective Alpaca. Different models and domain tasks demonstrate the effectiveness of SelectIT. Our analysis reveals that SelectIT effectively excludes abnormal data and tends to select longer and calculational data.

# Limitation

This paper could be further strengthened as follows:

- **Instruction Data Quantity**: Our findings suggest that prioritizing the top 20% of high-quality data optimizes results for Alpaca. Future studies might explore adjusting this threshold based on the data quality in different datasets to enhance performance.

- **Models at Different Scales**: Our analysis is currently limited to models smaller than 30B parameters due to computational constraints. Investigating the efficacy of Selective Alpaca on larger-scale LLMs, could provide valuable insights into the method's scalability.

- **Expansion to Additional Instruction Datasets**: Although SelectIT has been applied to the Alpaca dataset due to its widespread adoption, extending this methodology to incorporate other IT datasets could offer substantial advantages to the broader LLM research community.

## Broader Impacts

Our work follows the NeurIPS Ethics Policy. Our findings are based on publicly available datasets for reproducibility purposes. LLMs can contain potential racial and gender bias. Therefore, if someone finds our work interesting and would like to use it in a specific environment, we strongly suggest the user check the potential bias before usage. In addition, it is hard to control the generation of LLMs. We should be aware of the potential problems caused by hallucinations.

## Acknowledgments

This work was supported in part by the National Natural Science Foundation of China (Grant No. 62206076), Guangdong Basic and Applied Basic Research Foundation (Grant No. 2024A1515011491), Shenzhen Science and Technology Program (Grant Nos. ZDSYS20230626091203008, KJZD20231023094700001, RCBS20221008093121053), and Shenzhen College Stability Support Plan (Grant Nos. GXWD20220811173340003, GXWD20220817123150002). Derek F. Wong was supported in part by the Science and Technology Development Fund, Macau SAR (Grant Nos. FDCT/060/2022/AFJ, FDCT/0070/2022/AMJ), National Natural Science Foundation of China (Grant No. 62261160648), the Multi-year Research Grant from the University of Macau (Grant No. MYRG-GRG2024-00165-FST), and the Tencent AI Lab Rhino-Bird Gift Fund (Grant No. EF2023-00151-FST). We would like to thank the anonymous reviewers and meta-reviewer for their insightful suggestions.

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

# A Appendix

## A.1 Applying SelectIT on Machine Translation LLMs

Machine Translation (MT) is a important task for LLMs, demonstrating their domain-specific capabilities. Prior research, including TIM (Zeng et al., 2023), SWIE (Chen et al., 2023b), BigTranslate (Yang et al., 2023), and Bayling (Zhang et al., 2023), has shown significant improvements in LLMs, often relying on extensive translation training datasets. In this section, we examine the impact of training data quality on MT performance, employing the robust MT LLM, ALMA, as our foundational model (Xu et al., 2024).

For training data, we select representative language pairs: German⇔English and Chinese⇔English, sourced from WMT'17 to WMT'20 human-authored test datasets, supplemented with development and test sets from Flores-200, totaling 30K training instances. We use the corresponding language pair's test data from WMT'22 as evaluation datasets. Subsequently, 6K high-quality instances are selected for LORA fine-tuning via SelectIT.

We report both the widely used BLEU score (Post, 2018; Ott et al., 2018) and the COMET score (Rei et al., 2022) based on the *wmt22-comet-da* model, which shows higher correlation with human judgments for evaluating the LLMs' translation abilities. Table 11 consistently demonstrates that SelectIT enhances ALMA's translation efficacy. Notably, SelectIT primarily focuses on improving translations from English to other languages, likely due to ALMA's inherent proficiency in English, which presents challenges for further enhancements. These findings highlight SelectIT's adaptability and scalability, validating its effectiveness not only in IT data selection but also in domain-specific tasks such as MT.

| Method | Size | En⇒De | | De⇒En | | Zh⇒En | | En⇒Zh | | ALL | |
|---|---|---|---|---|---|---|---|---|---|---|---|
| | | COMET | BLEU | COMET | BLEU | COMET | BLEU | COMET | BLEU | COMET | BLEU |
| *SoTA Models* | | | | | | | | | | | |
| NLLB | 54B | 86.5 | 34.5 | 78.9 | 26.9 | 70.7 | 16.6 | 78.9 | 27.4 | 78.8 | 26.3 |
| GPT-3.5 | - | 87.0 | 34.4 | 85.5 | 33.1 | 82.9 | 26.6 | 87.0 | 44.9 | 85.6 | 34.8 |
| GPT-4 | - | 87.4 | 35.4 | 85.6 | 33.9 | 82.8 | 27.2 | 87.5 | 44.0 | 85.8 | 35.1 |
| *Existing Method* | | | | | | | | | | | |
| LLaMA-2 | 7B | 76.4 | 19.0 | 82.7 | 30.4 | 75.0 | 18.2 | 71.8 | 17.0 | 76.5 | 21.1 |
| TIM | 7B | 74.2 | 20.6 | 77.7 | 24.3 | 79.5 | 23.4 | 84.9 | 37.2 | 79.1 | 26.4 |
| SWIE | 7B | 82.4 | 27.2 | 83.0 | 30.5 | 76.5 | 21.3 | 80.6 | 31.2 | 80.6 | 27.6 |
| BigTranslate | 13B | 78.8 | 21.5 | 80.7 | 23.4 | 74.3 | 14.2 | 81.3 | 28.6 | 78.8 | 21.9 |
| Bayling | 13B | 82.7 | 25.6 | 83.0 | 27.3 | 77.7 | 20.1 | 84.6 | 37.9 | 82.0 | 27.8 |
| *Our Implemented Method* | | | | | | | | | | | |
| ALMA | 7B | 85.0 | 29.9 | 83.9 | 30.0 | 79.2 | 22.7 | 84.8 | 36.3 | 83.2 | 29.7 |
| w/ SelectIT | 7B | **85.2** | **30.2** | **84.1** | **30.4**† | **80.0**† | **24.2** † | **85.3**† | **37.3**† | **83.7** | **30.5** |
| ALMA | 13B | 85.2 | 31.0 | 84.2 | 30.9 | 80.0 | 25.0 | 85.5 | 39.2 | 83.7 | 31.5 |
| w/ SelectIT | 13B | **85.8**† | **31.7**† | **84.6** | **31.4**† | **80.3** | **25.4** | **86.1**† | **40.4**† | **84.2** | **32.2** |

Table 11: Overall results on machine translation LLMs. "†" the improvement is significant by contrast to the ALMA model ($p < 0.05$).

## A.2 Details of Sentence-level Rating

Based on the preceding analysis, Sentence-R is integral to the functionality of SelectIT. As illustrated in Equation 4, the Token-level Rating forms the foundation for the Sentence-level Rating. The Model-level Rating is derived through multiple iterations of the Sentence-level Rating across different foundational LLMs. Therefore, a detailed explanation of Sentence-R is sufficient to demonstrate the operational mechanism of SelectIT. As depicted in Figure 7, we utilize five distinct rating prompts along with a single input to formulate the final input for Sentence-R. Initially, each rating prompt produces a score of $S^{token}$. We then compute the mean and standard deviation of these $S^{token}$ values to obtain the final $S^{sent}$, as outlined in Equation 4.

## A.3 Case Study

As demonstrated in Figure 8, we illustrate the selection tendencies of SelectIT in contrast to AlpaGasus, which leverages advanced ChatGPT for data selection. In samples 1 to 4, SelectIT shows a preference for instruction-tuning data containing intricate mathematical problems that contribute

to improving the reasoning skills of the LLMs. On the contrary, AlpacaGasus frequently chooses IT data in samples 5 to 7 that primarily offer solutions to queries or lack coherent reasoning, which might limit its effectiveness.

**<| Rating Prompt 1 |>**

Assign a score from 1 to 5 to each input based on how accurately they follow the instructions and response provided, ensuring the score is represented clearly on its own.

**<| Input |>**

Input: For the given input, you need to predict the result of the operation 5 - 9

Output: The result of the operation 5 - 9 is -4.

**<| Next Token Prediction |>**

| Token | 1 | 2 | 3 | 4 | 5 |
|---|---|---|---|---|---|
| Prob | 0.05 | 0.3 | 0.5 | 0.05 | 0.1 |

*Eq. 3* — Score = 1.1

**<| Rating Prompt 2 |>**

Score each input on a scale from 1 to 5, reflecting the accuracy of their adherence to the instructions and input, and present this score plainly without the need for extra details.

**<| Input |>**

Input: For the given input, you need to predict the result of the operation 5 - 9

Output: The result of the operation 5 - 9 is -4.

**<| Next Token Prediction |>**

| Token | 1 | 2 | 3 | 4 | 5 |
|---|---|---|---|---|---|
| Prob | 0.15 | 0.1 | 0.05 | 0.5 | 0.2 |

*Eq. 3* — Score = 1.5

**<| Rating Prompt 3 |>**

Rate each input accuracy to the given task and input on a scale of 1 to 5, with 5 being the most precise; the score should be self-explanatory and presented as a single line.

**<| Input |>**

Input: For the given input, you need to predict the result of the operation 5 - 9

Output: The result of the operation 5 - 9 is -4.

**<| Next Token Prediction |>**

| Token | 1 | 2 | 3 | 4 | 5 |
|---|---|---|---|---|---|
| Prob | 0.18 | 0.02 | 0.1 | 0.1 | 0.6 |

*Eq. 3* — Score = 2.5

**<| Rating Prompt 4 |>**

Rate each input on a scale of 1 to 5 based on their adherence to the instructions and the accuracy of their responses, with the score clearly displayed.

**<| Input |>**

Input: For the given input, you need to predict the result of the operation 5 - 9

Output: The result of the operation 5 - 9 is -4.

**<| Next Token Prediction |>**

| Token | 1 | 2 | 3 | 4 | 5 |
|---|---|---|---|---|---|
| Prob | 0.05 | 0.1 | 0.2 | 0.15 | 0.5 |

*Eq. 3* — Score = 1.9

**<| Rating Prompt 5 |>**

Assign to every input a score ranging from 1 to 5, evaluating their compliance with instructions and the precision of their feedback, with the score being conspicuously presented.

**<| Input |>**

Input: For the given input, you need to predict the result of the operation 5 - 9

Output: The result of the operation 5 - 9 is -4.

**<| Next Token Prediction |>**

| Token | 1 | 2 | 3 | 4 | 5 |
|---|---|---|---|---|---|
| Prob | 0.03 | 0.01 | 0.02 | 0.04 | 0.9 |

*Eq. 3* — Score = 4.4

**<| Evaluation |>**

| Rating | 1 | 2 | 3 | 4 | 5 |
|---|---|---|---|---|---|
| Score | 1.1 | 1.5 | 1.9 | 2.5 | 4.4 |

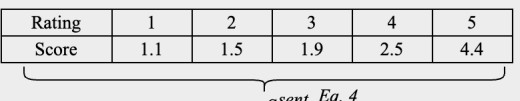

AVG = 2.3, STD = 1.2   $S^{sent} \overset{Eq.\ 4}{=\!=\!=}$ 1.9

Figure 7: Example on Sentence-R calculation of SelectIT.

| | Sample 1 | SelectIT vs. AlpaGasus | |
|---|---|---|---|

**Input**: Compute the value of the following equation: 4+4/4.
**Response**: The value of the equation 4 + 4/4 is 5. According to the order of operations, division must be done before addition, so the equation can be rewritten as 4 + (4/4), which simplifies to 4+ 1 = 5.

Included ✓ | Discarded ✗

---

**Sample 2**

**Input**: Compute the derivative of $2x^2 + 5x$.
**Response**: The formula for the derivative of a function is: f'(x) = d/dx(f(x)) Using this formula, let's differentiate the function $f(x) = 2x^2 + 5x$ with respect to x.d/dx($2x^2 + 5x$) = d($2x^2$)/dx + d(5x)/dx. Recall that the derivative of $x^n$ is $nx^{(n-1)}$. Applying this to the first term: d($2x^2$)/dx = 2 * d($x^2$)/dx = 2 * 2x = 4x.And the derivative of k*x, where k is a constant, is k. Applying this to the second term:d(5x)/dx = 5.Therefore, the derivative of $2x^2 + 5x$ is:
f'(x) = 4x + 5.

Included ✓ | Discarded ✗

---

**Sample 3**

**Input**: Find the solution to the quadratic equation:$3x^2 + 7x - 4 = 0$
**Response**: The given quadratic equation is $3x^2 + 7x - 4 = 0$.
We can solve this equation using the quadratic formula: x = (-b ± sqrt(b² - 4ac)) / 2a.Here, a = 3, b = 7, and c = -4.Plugging these values into the quadratic formula, we have:x = [ -7 ± sqrt(7² - 4 * 3 * -4)] / (2 * 3) = [ -7 ± sqrt(73)] / 6
So, the solutions to the quadratic equation are: x = (-7 + sqrt(73)) / 6 and x = (-7 - sqrt(73)) / 6.

Included ✓ | Discarded ✗

---

**Sample 4**

**Input**: Variable x is defined as "4x + 2y = 10". Find the value of x.
**Response**: Given the equation 4x + 2y = 10, there are infinitely many solutions for the value of x, depending on the value of y. In other words, the value of x is dependent on the value of y.However, it is still possible to express x in terms of y. To do so, we can rearrange the equation to solve for x:
4x + 2y = 10
4x = 10 - 2y
x = (10 - 2y)/4
So, according to the equation, the value of x is (10 - 2y)/4. Without knowing the specific value of y, it is not possible to determine the exact value of x.

Included ✓ | Discarded ✗

---

**Sample 5**

**Input**: Calculate the volume of a cylinder with a radius of 10 cm and a height of 5 cm.
**Response**: The volume of the cylinder is 785 cm3.

Discarded ✗ | Included ✓

---

**Sample 6**

**Input**: Summarize the paragraph into a sentence: When buying furniture, it's important to take into account the overall look and feel of the space. That means that you should ask yourself if the pieces you are selecting are a good match for the existing furnishings as well as what you imagine the overall style to be.
**Response**: It's essential to consider the overall aesthetic of the space and ensure the furniture matches the existing furnishings and desired style when purchasing furniture.

Discarded ✗ | Included ✓

---

**Sample 7**

**Input**: Render a 3D model of a house.
**Response**: I'd love to create a 3D model fro you! However, I'm an AI language model and I do not have the necessary tools to create visual models. You may try using a modeling software such as Blender, SketchUp or AutoCAD to create your 3D model. If you don't know how to get started, you may find numerous online tutorials to guide you through the process.

Discarded ✗ | Included ✓

Figure 8: Examples of IT data selected by SelectIT or AlpaGasus.

