# OpenReview forum: "SelectIT: Selective Instruction Tuning for LLMs via Uncertainty-Aware Self-Reflection"
_NeurIPS.cc/2024/Conference — NeurIPS 2024 poster_

### Official Review · Reviewer_4rCk · 2024-06-12

**Soundness:** 3
**Presentation:** 2
**Contribution:** 3
**Rating:** 5
**Confidence:** 4

**Summary:**

The paper introduces SelectIT, a novel instruction tuning (IT) method that leverages the intrinsic uncertainty in large language models (LLMs) to select high-quality IT data without requiring additional resources. The authors present a curated IT dataset, Selective Alpaca, derived from the Alpaca-GPT4 dataset using SelectIT. The empirical results demonstrate significant improvements in model performance across various tasks. The paper highlights the robustness and efficiency of SelectIT in enhancing the capabilities of LLMs.

**Strengths:**

-  The approach of using the intrinsic uncertainty of LLMs for IT data selection is innovative and eliminates the need for external resources, making it cost-effective and widely applicable.
-  The submission is technically sound, with well-supported claims through theoretical analysis and comprehensive experimental results.

**Weaknesses:**

- Figure 2 is not clear to me. I suggest giving a brief description of each selection stage in the caption for easy understanding.
- It seems like the foundational LLM's capacity becomes the key for this data selection. In this case, the selection method may not be as effective as other external resource methods.

**Questions:**

- How does SelectIT perform when applied to instruction tuning datasets other than Alpaca-GPT4?
- Can the authors provide more detailed examples or case studies illustrating the differences between data selected by SelectIT and other methods? And how these different influence the final model performance?
- How does the performance of SelectIT change with different values of the uncertainty factor (α) ?

**Limitations:**

Yes

---

> ### Author Rebuttal · Authors · 2024-08-07
>
> Thank you for your insightful review.
>
> >Q1: Figure 2 is not clear to me. I suggest giving a brief description of each selection stage in the caption for easy understanding.
>
> Thank you for your advice. Following your suggestion, we will revise the caption of Figure 2 to: `Overall Framework of Our Proposed SelectIT Method. In the Token-level Self-Reflection, we employ the foundation model to rate the IT data from 1 to $K$. For Sentence-level Self-Reflection, we leverage the uncertainty of varied prompts on LLMs to enhance the rating process. In the Model-level Self-Reflection, we harness uncertainty among different LLMs to facilitate a collaborative decision-making process in selecting IT data. Finally, different levels of self-reflection are reasonably combined into SelectIT, which can effectively select high-quality instruction tuning data without relying on additional resources.`
>
> >Q2:  It seems like the foundational LLM's capacity becomes the key for this data selection. In this case, the selection method may not be as effective as other external resource methods.
>
> - **Comparison with other external resource methods:**
> In Table 1, we compare SelectIT with various other external resource methods such as AlpaGasus [1], Q2Q [2], and Instruction Mining [3]. SelectIT demonstrates superiority in the abundant experiments including six different tasks and varying model parameter sizes. Furthermore, as shown in Table 7, SelectIT requires less time and fewer resources compared to AlpaGasus or its variant. Therefore, we have grounds to believe that leveraging the capacity of LLMs without external resources is sufficient for selecting high-quality samples.
>
> - **Emergence of more powerful LLMs:**  Based on the main result of Table 1, the LLaMA-2 series base models we used have exhibited sufficient capability in selecting higher-quality samples. With the continuous development of our LLM community, more powerful open-source LLMs are emerging, such as LLaMA-3.1, Mistral-7B-v0.2, and Qwen2 etc. Given that the performance of SelectIT is intrinsically connected to the capacities of foundation LLMs, its efficiency is not only unrestricted but also anticipated to enhance concurrently with the advancement of LLMs in the future.
>
> [1] AlpaGasus: Training A Better Alpaca with Fewer Data. ICLR, 2024
>
> [2] From Quantity to Quality: Boosting LLM Performance with Self-Guided Data Selection for Instruction Tuning. NAACL, 2024
>
> [3] Instruction Mining: Instruction Data Selection for Tuning Large Language Models. arXiv, 2023.
>
> >Q3:  How does SelectIT perform when applied to instruction tuning datasets other than Alpaca-GPT4?
>
> **In Section 5.1 of "Various Domain-specific Tasks", SelectIT is also applied to different datasets such as machine translation datasets.** As illustrated from Lines  267-272 in the paper, following the configuration of ALMA[1], we opt for a total of 30K training examples for representative language pairs and utilize SelectIT to select a subset of 6K data for comparative training. As Table 6 demonstrates, SelectIT can invariably boost the translation performance of ALMA [1] using fewer yet higher-quality training data.
>
> **To further alleviate concerns, we apply SelectIT to other instruction tuning datasets such as WizardLM and larger Orca-GPT4,** while keeping the hyper-parameters consistent. For more details, please refer to the Q1 of General Response.
>
> [1] A Paradigm Shift in Machine Translation: Boosting Translation Performance of Large Language Models. ICLR, 2024.
>
>
> >Q4 :  Can the authors provide more detailed examples or case studies illustrating the differences between data selected by SelectIT and other methods? And how these different influence the final model performance?
>
> In Figure 5, we have comprehensively discussed the differences in the samples selected by SelectIT and other methods such as AlpaGasus.
>
> - **Length**: Relative to AlpaGasus and the full set of Alpaca, the average sample length of SelectIT is the largest, reaching 241.85. Length is a crucial dimension in assessing sample quality [1]. Longer sentences typically contain more information or procedural steps, which are beneficial in enhancing model capabilities.
>
> - **Calculation**: As shown in Lines 61-62 and Figures 5-6,  one of the primary contributions of SelectIT is that it prefers additional calculation-type IT data which enhances the LLMs' reasoning capabilities, offering a new perspective on the characteristics of optimal IT data. As demonstrated in Table 1, SelectIT achieved optimal results on benchmarks assessing computational and mathematical capabilities. The enhancement in model capabilities further improved the LLMs' coding abilities, ultimately leading to an overall enhancement in comprehensive capabilities.
>
> Furthermore, in Appendix Section A.4, we have presented relevant case studies contrasting SelecIT and AlpaGasus. In samples 1 to 4 of Figure 8, SelectIT demonstrates a preference for instruction-tuning data featuring complex mathematical problems, aligning with the findings of Section 5.4. In contrast, AlpacaGasus usually selects IT data predominantly providing solutions to queries or lacking coherent reasoning in samples 5 to 7, potentially hindering its effectiveness.
>
> [1] Long Is More for Alignment: A Simple but Tough-to-Beat Baseline for Instruction Fine-Tuning. arXiv, 2024.
>
> >Q5:  How does the performance of SelectIT change with different values of the uncertainty factor (α) ?
>
> Please refer to the Q3 of General Response.

---

> > ### Comment · Reviewer_4rCk · 2024-08-13
> >
> > Thanks for your response. Regarding the 'Length' of the selected data, I cannot be convinced that one of the primary contributions is that a longer instruction indicates better quality. LLMs have a verbosity bias, preferring longer output even when it's not necessarily clearer [1]. Therefore, I will maintain my score.
> >
> > [1] ZHng et al. Judging LLM-as-a-Judge with MT-Bench and Chatbot Arena

---

> ### Author Response · Authors · 2024-08-10
> **Look forward to follow-up discussion.**
>
> Dear Reviewer 4rCk,
>
> We greatly appreciate your valuable feedback and suggestions! To address your concerns, we have run extensive experiments on various datasets and made additional insightful observations. The results will be a nice plus to improve the clarity and useability of our method.
>
> As the discussion period is coming to an end, we kindly request that you review these results. If our results and explanations help address your concerns, we would be grateful if you could acknowledge our rebuttal and consider adjusting your score accordingly.
>
> Best wishes,
> The Authors

---

> ### Author Response · Authors · 2024-08-13
> **Request to review the rebuttal [Author-Reviewer discussion phase ending soon]**
>
> Dear Reviewer 4rCk,
>
> Thank you for your time reviewing the paper. Your constructive feedback will help improve the quality of our paper.
>
> We have also addressed all your concerns in our rebuttal point-by-point. As we are towards the end of the author-reviewer discussion period, we request you to please go through our rebuttal, and we would be more than happy to address any more concerns you have.
>
> We thank you again for your time!
>
> Best,
> The Authors

---

> > ### Comment · Area_Chair_UByw · 2024-08-13
> >
> > Dear reviewer 4rCk
> >
> > Could you please take a look at the responses of the authors and let us know your thoughts on them? Are you satisfied with the responses and do you have some updates on your comments? Please also take a look at other reviews and share with us your thoughts on whether the paper is ready for publication. We need your response so that we can make an informed decision on this paper.
> >
> > AC

---

> ### Author Response · Authors · 2024-08-14
> **Response to follow-up questions.**
>
> Thank you for your response.
>
> We will give a clearer explanation of SelectIT's preference for longer instruction data.
>
> As shown in the "Introduction" section and "Insights of Selective Data Curation" section, the core feature of Selective Alpaca is the increase in the proportion of computational data, which is different from other methods. Computational data is often longer than other types of data because it requires more reasoning steps, which further leads to the longer overall length of Selective Alpca. So, "Length" may still be the important and easy-to-get feature for evaluating the quality of sample data [1-4].
>
> Secondly, SelectIT does not simply prefer longer data. As shown in Table 8 of our paper (also the table below), we compared the SelectIT with the previous work [1], which selects samples only by length. The results show that SelectIT can achieve better results with the same amount of data.
>
> |                        | LLaMA-2-7B | LLaMA-2-13B | ALMA-7B | ALMA-13B |
> |:----------------------:|:----------:|:-----------:|---------|----------|
> |      Alpace-GPT4       |    34.1    |    44.2     | 29.7    | 31.5     |
> |  Alpace-GPT4 + Length  |    35.5    |    47.1     | 30.1    | 31.8     |
> | Alpace-GPT4 + SelectIT |    **36.2**    |    47.1     | **30.5**    | **32.2**     |
>
> Finally, we further compared the output lengths of models trained on Selective Alpaca and Alpace-GPT4 on the Alpaca-eval benchmark. As shown in the following table, with the output lengths being similar, Selective Alpaca achieves a 1.5% improvement, which effectively eliminates the bias in sample length and explanation of the doubts about the preference of LLMs for samples with longer length.
>
> |                  | Length | Win Rate |
> |:----------------:|:------:|:----:|
> |   Alpace-GPT4    |  807   | 34.2 |
> | Selective Alpaca |  815   | 35.7 |
>
> Thank you again for taking the time to read our response! We are more than happy to continue the discussion if you are still confused about the paper.
>
> [1] Long Is More for Alignment: A Simple but Tough-to-Beat Baseline for Instruction Fine-Tuning. arXiv, 2024.
> [2] Zero-Shot Generalization during Instruction Tuning: Insights from Similarity and Granularity. arXiv, 2024.
> [3] LIMA: Less Is More for Alignment. NeurIPS, 2023.
> [4] AlpaGasus: Training A Better Alpaca with Fewer Data. ICLR, 2024.

---

### Official Review · Reviewer_CKC2 · 2024-07-12

**Soundness:** 3
**Presentation:** 3
**Contribution:** 2
**Rating:** 5
**Confidence:** 4

**Summary:**

In this paper, authors leverage LLMs to score the training data in terms of the token and sentence and model. They train various models on their select models to show the effectiveness on various benchmarks. Their main contribution is to apply foundations to judge the quality of training data from different granularity without any external resources.

**Strengths:**

1. Their idea is easy to understand and the writing is pretty clear to me.
2. They conduct a lot of experiments on several benchmarks to show the effectiveness.

**Weaknesses:**

1. From the table 3, it seems sentence-R works the best so what if we trained model on the data just from the sentence-R?  BTW, what is the ID 6, 7, 8 means in that Table?
2. My main question is that do we really need all three types selection models, can one or two of them already achieve great performance?
3. What is the overhead of this data selection model? Like how long does it take to finish the process on 10K dataset.

**Questions:**

Please refer to the  above.

**Limitations:**

Yes

---

> ### Author Rebuttal · Authors · 2024-08-07
>
> Thank you for your insightful review.
>
> >Q1: From the table 3, it seems Sentece-R works the best so what if we trained model on the data just from the Sentece-R? BTW, what is the ID 6, 7, 8 means in that Table?
>
> Thank you for your suggestion. Actually, in Table 1, we have provided the experiment that trained the model on the data from Sentece-R, referring to System 7 and System 16. The results in Table 1 indicate that Sentece-R achieved the best outcomes among three different level methods (Token-R, Sentece-R, Model-R) of SelectIT,  which aligns with the conclusions in Table 3 and further dissects the key aspects of SelectIT performance.
>
> Additionally, IDs 6, 7, and 8 in Table 3 correspond to the system of the same IDs in Table 1, which means using the same methods to select data. We will provide clearer explanations for Table 3 in the next version.
>
> >Q2: My main question is that do we really need all three types selection models (method), can one or two of them already achieve great performance?
>
>
> There might be an ambiguity that the three types are either the three different levels of self-reflection strategies or the three different bases models, we will address both aspects as follows.
>
> - **Different Levels of Self-Reflection Strategies**:
> Based on Table 1 and Table 3, Sentence-R performs better than other select methods, such as Token-R and Model-R,  which is attributed to the use of different prompts to quantify the uncertainty of the LLMs. So the single method, Sentence-R is good enough to select data. Furthermore, we integrate various levels of self-reflection: token, sentence, and model, to form the final SelectIT, achieving the best performance.
>
> - **Different Types of Bases Models**:
> As you expect, SelectIT still performs well when there are only one or two base models. Based on Table 1 and Table 3, Sentence-R plays the most important role in SelectIT, which only uses the single base model. Model-R, as an extension of SelectIT, allocates weights to the result of each model based on the number of model parameters for data selection, mainly enhancing the robustness of the algorithm.
>
> Additionally, we will supplement the experiments which using varying numbers of models to select the  instruction data, with the results presented below:
>
> |                    Used Models          | MMLU | BBH  |   GSM   | Tydiqa | CodeX | AE   |     AVG     |
> |:--------------------------------------:|:----:|:----:|:-------:|:------:|------|------|:-----------:|
> |                   None (Baseline)                    | 46.5 | 38.4 |  15.0   |  43.4  | 26.8 | 34.2 |    34.1     |
> |               LLaMA-2-7B               | 47.6 | 39.1 |  15.0   |  46.1  | 28.2 | 35.2 |    35.2     |
> |        LLaMA-2-7B + LLaMA-2-13B        | 47.7 | 38.1 |  15.5   |  47.8  | 28.4 | 35.1 |    35.4     |
> | LLaMA-2-7B + LLaMA-2-13B + LLaMA-2-70B | 47.4 | 40.6 |  16.8   |  47.4  | 29.4 | 35.7 |  **36.2**   |
>
> Based on the above analysis and experimental results, it is evident that SelectIT still works well when there is a single base model, referred to as the Sentece-R. More base models are also beneficial for selecting high-quality data, which can effectively enhance the robustness of SelectIT.
>
>
> >Q3:  What is the overhead of this data selection model? Like how long does it take to finish the process on 10K dataset.
>
> In Section 5.3, we have elaborated on and compared the efficiency of SelectIT with other methods. Based on the Alpaca dataset, we deployed 4 A800 80G GPUs to select high-quality IT data, which ultimately required **5.8 hours**. In contrast, utilizing the ChatGPT API required significantly more time, **19.07 hours**. This issue is even more pronounced with the more powerful GPT-4, demonstrating the practical utility of SelectIT. For the 10K dataset, keeping the same GPU devices as above, it would necessitate 1.2 hours.

---

> > ### Comment · Reviewer_CKC2 · 2024-08-13
> >
> > Thanks for the response and I am thinking maybe we still need to understand why combination of models can yield better results on some benchmarks, for example, we do not see meaningful improvement on MMLU.  I will keep my score. thanks!

---

> ### Author Response · Authors · 2024-08-10
> **Look forward to follow-up discussion.**
>
> Dear Reviewer CKC2,
>
> We greatly appreciate your valuable feedback and suggestions! To address your concerns, we have run extensive experiments on various datasets and made additional insightful observations. The results provide more evidence for our method justification.
>
> As the discussion period is coming to an end, we kindly request that you review these results. If our results and explanations help address your concerns, we would be grateful if you could acknowledge our rebuttal and consider adjusting your score accordingly.
>
> Best wishes,
> The Authors

---

> ### Author Response · Authors · 2024-08-13
> **Request to review the rebuttal [Author-Reviewer discussion phase ending soon]**
>
> Dear Reviewer CKC2,
>
> Thank you for your time reviewing the paper. Your constructive feedback will help improve the quality of our paper.
>
> We have also addressed all your concerns in our rebuttal point-by-point. As we are towards the end of the author-reviewer discussion period, we request you to please go through our rebuttal, and we would be more than happy to address any more concerns you have.
>
> We thank you again for your time!
>
> Best,
> The Authors

---

> > ### Comment · Area_Chair_UByw · 2024-08-13
> >
> > Dear reviewer CKC2
> >
> > Could you please take a look at the responses of the authors and let us know your thoughts on them? Are you satisfied with the responses and do you have some updates on your comments? Please also take a look at other reviews and share with us your thoughts on whether the paper is ready for publication. We need your response so that we can make an informed decision on this paper.
> >
> > AC

---

> ### Author Response · Authors · 2024-08-14
> **Response to follow-up questions.**
>
> Thank you for your response.
>
> For the Model-R of SelectIT, we used models to jointly determine high-quality data and reasonably weigh the sample quality according to the model parameters, which is similar to ensemble learning. As shown in "Insights of High-Quality Data in SelectIT" section and Figure 5, SelectIT which used multiple models is more conducive to screening data with biased calculation types. So, SelectIT can improve model reasoning ability and further improve the encoding ability of LLM, which ultimately leads to an overall improvement in comprehensive ability.
>
> Secondly, MMLU consists of a set of questions about 57 subjects and its multiple-choice format makes it suitable for probing models’ knowledge [1-2]. So it is often used to evaluate the foundation model competence, as LLMs mainly learn world knowledge in the pre-training stage, not the instruction tuning stage (only containing a few instances). So there is a slight improvement on MMLU which is relatively reasonable in SelectIT, and also consistent with the results of AlpaGasus [3].
>
> Thank you again for taking the time to read our response! We are more than happy to continue the discussion if you are still confused about the paper.
>
> [1] Measuring Massive Multitask Language Understanding. ICLR, 2021.
> [2] How Far Can Camels Go? Exploring the State of Instruction Tuning on Open Resources. NeurIPS, 2023.
> [3] AlpaGasus: Training A Better Alpaca with Fewer Data. ICLR, 2024.

---

### Official Review · Reviewer_6ofY · 2024-07-15

**Soundness:** 4
**Presentation:** 4
**Contribution:** 3
**Rating:** 7
**Confidence:** 4

**Summary:**

This paper presents SelectIT, a new way to improve how large language models (LLMs) follow instructions using their own uncertainty to pick better training data. This method is cost-effective as it doesn't require extra tools or data. The authors developed a new dataset called Selective Alpaca using SelectIT, which significantly improved LLMs' performance. The study suggests that using longer and more detailed data could be more beneficial for training models.

**Strengths:**

SelectIT is a bew and unique method that improves how large language models (LLMs) learn by using the model's own uncertainty to choose better training data. This approach is special because it doesn't need extra tools or data, making it less expensive and easier to use.

The paper includes detailed studies that show the method works well and supports its findings with strong evidence. SelectIT significantly improves how models perform compared to older methods, showing that it's a big improvement in teaching models.

The researchers made their methods and data public, allowing others to use and test them. This openness helps the scientific community to work together and trus the results more.

The paper is also written in a way that's easy to understand, which helps more people learn about SelectIT. It discusses how using longer and more detailed data can make models learn better. Additionally, the paper compares the costs of using SelectIT with using other common tools like the GPT-4 and ChatGPT APIs, showing that SelectIT can save money. This shows that SelectIT is not only effective but also cost-efficient for training models.

**Weaknesses:**

The paper states initially that it does not use extra resources for its new method. However, it mentions later that they used different sizes of LLaMA 2 models to help choose the best instruction data. This contradicts their first claim about not needing extra resources.

**Questions:**

"The paper states initially that it does not use extra resources for its new method. However, it mentions later that they used different sizes of LLaMA 2 models to help choose the best instruction data. This contradicts their first claim about not needing extra resources. " Clarification?

Could you consider applying the SelectIT method to larger datasets, such as the Orca dataset for ChatGPT with 5 million samples and the GPT-4 dataset with 1 million samples? This would help to evaluate whether your method scales effectively with increased data volume.

**Limitations:**

the paper lacks a separate section on limitation of the approach.

---

> ### Author Rebuttal · Authors · 2024-08-07
>
> Thank you for your insightful review.
>
> >Q1:  The paper states initially that it does not use extra resources for its new method. However, it mentions later that they used different sizes of LLaMA 2 models to help choose the best instruction data. This contradicts their first claim about not needing extra resources.
>
> The 'Main Results' and 'Effect of Different Reflection Strategy' sections highlight the effectiveness and importance of Sentence-R in SelectIT. Serving as a highly adaptable method, SelectIT can handle varying quantities of base models. In instances where there is a single base model, SelectIT leverages Sentence-R for data selection without requiring additional models or data. However, when multiple base models are present, SelectIT appropriately allocates weights to the result of each model based on the number of model parameters for data selection, thereby boosting the algorithm's robustness.
>
> In summary, SelectIT possesses the ability to use the base model directly for data selection without needing extraneous resources and it could achieve comparable performance with only using one base model.
>
> >Q2:  Could you consider applying the SelectIT method to larger datasets, such as the Orca dataset for ChatGPT with 5 million samples and the GPT-4 dataset with 1 million samples? This would help to evaluate whether your method scales effectively with increased data volume.
>
> **In Section 5.1 of "Various Domain-specific Tasks", SelectIT has applied to different datasets such as machine translation datasets.** As illustrated from Lines 267-272 in the paper, following the configuration of ALMA [1], we opt for a total of 30K training examples for representative language pairs and utilize SelectIT to select a subset of 6K data for comparative training. As Table 6 demonstrates, SelectIT can invariably boost the translation performance of ALMA [1] using fewer yet higher-quality training data.
>
> **To further alleviate concerns, we apply SelectIT to other instruction datasets such as WizardLM and larger Orca-GPT4, while keeping the hyperparameters consistent.** The results are shown in the following table.
>
>
> |          | Data Size | MMLU | BBH  | GSM  | Tydiqa | CodeX | AE   | AVG  |
> |:---------------------:|------|:----:|:----:|:----:|:------:|-------|------|:----:|
> |       WizardLM   | 143,000     | 43.8 | 37.8 | 10.0 |  41.2  | 25.2  | 35.3 | 32.2 |
> | WizardLM w/ SelectIT |  28,600 | 45.1 | 40.1 | 11.0 |  43.1  | 27.5  | 34.7 | **33.6** |
> |       Orca-GPT4    |  994,895    | 40.1 | 35.6 | 13.0 |  46.0  | 23.3  | 38.1 | 32.7 |
> | Orca-GPT4 w/  SelectIT |198,979 | 43.9 | 38.7 | 16.5 |  42.0  | 27.7 | 37.4 | **34.4** |
>
> Based on various task types, including instruction tuning (IT),  machine translation (MT), and diverse instruction datasets such as Alpaca, WizardLM, and the larger Orca-GPT4, extensive experimental results have robustly validated the general applicability and robustness of SelectIT.
>
> [1] A Paradigm Shift in Machine Translation: Boosting Translation Performance of Large Language Models. ICLR, 2024.
>
>
> >Q3: The paper lacks a separate Section on the limitations of the approach.
>
> Sorry for the misunderstanding. We have already provided the "Limitations" Section in Appendix A.1 of our original submission. Later, we aim to integrate this section into the main text to provide a clearer understanding of SelectIT to audiences.

---

> > ### Comment · Reviewer_6ofY · 2024-08-09
> >
> > Thank you for addressing most of my concerns. I have increased the OA score.

---

### Official Review · Reviewer_KKrK · 2024-07-18

**Soundness:** 3
**Presentation:** 4
**Contribution:** 2
**Rating:** 6
**Confidence:** 4

**Summary:**

The paper introduces a data selection approach to instruction tuning (IT) for large language models (LLMs) by leveraging the intrinsic uncertainties within the models themselves. This method, termed SelectIT, aims to enhance IT data selection without the need for additional external resources, making it more cost-effective and accessible.

SelectIT utilizes three levels of uncertainty within LLMs—token-level, sentence-level, and model-level—to effectively rate and select high-quality IT data. The paper demonstrates the effectiveness of SelectIT through empirical results, showing improvements in model abilities on the open-instruct benchmark and other domain-specific tasks. Additionally, SelectIT is highlighted as a faster and more resource-efficient method compared to traditional approaches, making it a more practical solution for widespread adoption.

---------------------------------------------------------------------------------------------------------------------------------------------------------------
Thank you for your reply and I have updated my score accordingly.

**Strengths:**

**Originality**: The paper presents a data selection approach to instruction tuning by leveraging the intrinsic uncertainties within LLMs, without relying on additional fine-tuned models or API calls of ChatGPT or Claude.

**Quality**: The methodology has a clear explanation of the multi-granularity self-reflection technique. The empirical results are robust, demonstrating the effectiveness of SelectIT across various benchmarks and tasks.

**Clarity**: The paper is clearly written, with well-organized sections and comprehensive explanations of the proposed method and its implementation.

**Significance**: SelectIT addresses a critical limitation in the field of instruction tuning by removing the dependence on additional models or data, making the approach more cost-effective and scalable. The curated Selective Alpaca dataset offers performance improvements for LLMs, highlighting the practical impact of the research.

**Weaknesses:**

1. **Reliance on Tuning K**: The method depends on tuning the parameter \( K \), which can significantly affect model performance. This dependency introduces additional complexity and may require extensive experimentation to optimize.

2. **Use of Multiple Base Models**: Although the paper claims not to use external resources, it actually relies on multiple different base models to obtain the model-level score. This reliance contradicts the claim of being purely self-reflective and can be confusing, as the title suggests the method is entirely self-contained.

3. **Sentence-Level Quality Evaluation**: The sentence-level quality evaluation requires multiple rating prompts. It is unclear whether other methods, such as AlpaGasus, would achieve similar performance if they also used multiple rating prompts. This comparison is missing and could be critical for understanding the true advantage of the proposed method.

4. **Single Dataset Evaluation**: All evaluations are conducted on a single training dataset, making it questionable whether this method can be generalized to other datasets, such as WizardLM, or multi-turn data, like the ShareGPT dataset. Testing on a more diverse set of datasets would strengthen the paper's claims of robustness and general applicability.

**Questions:**

(1) How the α = 0.2 in the experiments is selected?
(2) Will optimal K be training dataset dependent?

**Limitations:**

See more details in Weaknesses section.

---

> ### Author Rebuttal · Authors · 2024-08-07
>
> Thank you for your insightful review.
>
> >Q1: Reliance on Tuning K: The method depends on tuning the parameter ( K ), which can significantly affect model performance. Will optimal K be training dataset dependent?
>
> The performance of SelectIT is not critically dependent on the parameter $K$, based on the conclusions of Section 5.1 "Effect of $K$ ". As illustrated in Figure 2, when $K$ is sufficiently large, such as 5, 7, or 9, the impact of parameter $K$ on LLMs performance is negligible, with performance fluctuations around just 0.3. This is due to the multi-level self-reflection mechanism, which effectively enhances the accuracy and stability of sample selection.
>
> Furthermore, based on Section 5.2, **we also apply the SelectIT to the representative domain-specific task, machine translation (MT),  which reuses the same hyperparameters $K$ and $\alpha$**. As shown in Table 6, we present outcomes for two distinct language pairs in four distinct directions and, SelectIT can consistently enhance the translation performance of ALMA [1],  which is widely recognized as a leading machine translation LLM.
>
> Finally, referring to the response to Q1 of General Response, we also apply the same hyper-parameters of $K$ on the other datasets such as the WizardLM and larger Orca. The results demonstrate that the same $K$ hyper-parameters of SelectIT still yield consistently strong performance across different datasets. Therefore, extensive experimentation is not required to adjust K hyper-parameters, and it is independent of the training dataset.
>
> [1] A Paradigm Shift in Machine Translation: Boosting Translation Performance of Large Language Models. ICLR, 2024.
>
>
> >Q2: Use of Multiple Base Models: Although the paper claims not to use external resources, it actually relies on multiple different base models to obtain the model-level score. This reliance contradicts the claim of being purely self-reflective and can be confusing, as the title suggests the method is entirely self-contained.
>
> Please refer to the Q2 of General Response.
>
> >Q3:  Sentence-Level Quality Evaluation: The sentence-level quality evaluation requires multiple rating prompts. It is unclear whether other methods, such as AlpaGasus, would achieve similar performance if they also used multiple rating prompts. This comparison is missing and could be critical for understanding the true advantage of the proposed method.
>
> Thank you for your advice. We will provide a clearer explanation of Sentence-Level Quality Evaluation. To the best of our knowledge, **we are the first one to use the effect of different prompts on LLMs to select high-quality data.** Following Figure 2 and Equation 4, we do more than use K times prompting and averaging the rating score. More importantly, we calculated the standard deviation of the rating score to quantify the model's uncertainty, which is not only the first attempt in our work but also an important contribution to our work.
>
> For the previous techniques, we compare Selective Alpaca with the official open-source dataset [1] in Table 1, which we think is fair enough. **Additionally, based on the resource consumption we reported in Table 7 of the efficiency of SelectIT, using ChatGPT to K times-ly selects IT data is significant consumption and challenging to implement, which is also proved by recent research** [2].
>
> [1] https://github.com/gpt4life/alpagasus
>
> [2] Liu et al., What Makes Good Data for Alignment? A Comprehensive Study of Automatic Data Selection in Instruction Tuning. ICLR 2024.
>
> >Q4: Single Dataset Evaluation: All evaluations are conducted on a single training dataset, making it questionable whether this method can be generalized to other datasets, such as WizardLM, or multi-turn data, like the ShareGPT dataset. Testing on a more diverse set of datasets would strengthen the paper's claims of robustness and general applicability.
>
> Sorry for the misunderstanding. **In Section 5.1 of "Various Domain-specific Tasks", SelectIT has applied to different datasets such as MT datasets.** As illustrated from Lines 267-272 in the paper, following the configuration of ALMA [1], we opt for a total of 30K training examples for representative language pairs and utilize SelectIT to select a subset of 6K data for comparative training. As Table 6 demonstrates, SelectIT can invariably boost the translation performance of ALMA [1] using fewer yet higher-quality training data.
>
> **To further alleviate concerns, we apply SelectIT to other instruction datasets such as WizardLM and larger Orca-GPT4.** For more details, please refer to the Q1 of General Response.
>
> [1] A Paradigm Shift in Machine Translation: Boosting Translation Performance of Large Language Models. ICLR, 2024.
>
> >Q5: How the $\alpha$ = 0.2 in the experiments is selected?
>
> The value of $\alpha$ is determined through our analytical research on Sentence-level Self-Reflection. To further justify the appropriateness of $\alpha$ = 0.2, we will conduct supplementary experiments on varying α values of  SelectIT on the Alpaca datasets. For more details, please refer to the Q3 of General Response.

---

> ### Author Response · Authors · 2024-08-10
> **Look forward to follow-up discussion.**
>
> Dear Reviewer KKrK,
>
> We greatly appreciate your valuable feedback and suggestions! To address your concerns, we have run extensive experiments on various datasets and made additional insightful observations. The results further confirm the generality of our proposed method.
>
> As the discussion period is coming to an end, we kindly request that you review these results. If our results and explanations help address your concerns, we would be grateful if you could acknowledge our rebuttal and consider adjusting your score accordingly.
>
> Best wishes,
> The Authors

---

> ### Author Response · Authors · 2024-08-13
> **Request to review the rebuttal [Author-Reviewer discussion phase ending soon]**
>
> Dear Reviewer KKrK,
>
> Thank you for your time reviewing the paper. Your constructive feedback will help improve the quality of our paper.
>
> We have also addressed all your concerns in our rebuttal point-by-point. As we are towards the end of the author-reviewer discussion period, we request you to please go through our rebuttal, and we would be more than happy to address any more concerns you have.
>
> We thank you again for your time!
>
> Best,
> The Authors

---

> > ### Comment · Area_Chair_UByw · 2024-08-13
> >
> > Dear reviewer KKrK
> >
> > Could you please take a look at the responses of the authors and let us know your thoughts on them? Are you satisfied with the responses and do you have some updates on your comments? Please also take a look at other reviews and share with us your thoughts on whether the paper is ready for publication. We need your response so that we can make an informed decision on this paper.
> >
> > AC

---

### Author Rebuttal · Authors · 2024-08-07

We thank the reviewers for the insightful comments and constructive suggestions, which will serve to improve the paper considerably. We will attend to all comments to the best extent.

> Q1: Applying the SelectIT on other representative instruction tuning datasets.

To further demonstrate the robustness of SelectIT, we apply it to two additional commonly used datasets: WizardLM and Orca-GPT4. **At the same time, we maintain consistent hyperparameters such as $\alpha$ and $K$ and use the LLaMA-2-7B as our base model.**  The results are presented in the following table.

|          | Data Size | MMLU | BBH  | GSM  | Tydiqa | CodeX | AE   | AVG  |
|:---------------------:|------|:----:|:----:|:----:|:------:|-------|------|:----:|
|       WizardLM   | 143,000     | 43.8 | 37.8 | 10.0 |  41.2  | 25.2  | 35.3 | 32.2 |
| WizardLM w/ SelectIT |  28,600 | 45.1 | 40.1 | 11.0 |  43.1  | 27.5  | 34.7 | **33.6** |
|       Orca-GPT4    |  994,895    | 40.1 | 35.6 | 13.0 |  46.0  | 23.3  | 38.1 | 32.7 |
| Orca-GPT4 w/  SelectIT |198,979 | 43.9 | 38.7 | 16.5 |  42.0  | 27.7 | 37.4 | **34.4** |

Based on three diverse instruction datasets such as Alpaca-GPT4, WizardLM, and the larger Orca-GPT4, extensive experimental results robustly validate the general applicability and robustness of SelectIT.

> Q2: The use of extra models.

The 'Main Results' and 'Effect of Different Reflection Strategy' sections highlight the effectiveness and importance of Sentence-R in SelectIT. Serving as a highly adaptable method, SelectIT can handle varying quantities of base models. In instances where there is a single base model, SelectIT leverages Sentence-R for data selection without requiring additional models or data. However, when multiple base models are present, SelectIT appropriately allocates weights to the result of each model based on the number of model parameters for data selection, thereby boosting the algorithm's robustness.

In summary, SelectIT possesses the ability to use the base model directly for data selection without needing extraneous resources and it could achieve comparable performance with only using one base model.

> Q3:  How does the performance of SelectIT change with different values of the uncertainty factor (α)?

We will supplement the experiments about different $\alpha$ of SelectIT on the Alpaca datasets, which use the LLaMA-2-7B as the foundation model and keep other parameters unchanged.

|                 | MMLU | BBH  | GSM  | Tydiqa | CodeX | AE   |   AVG    |
|:---------------:|:----:|:----:|:----:|:------:|-------|------|:--------:|
| $\alpha$  = 0.2 | 47.4 | 40.6 | 16.8 |  47.4  | 29.4  | 35.7 | **36.2** |
| $\alpha$  = 0.4 | 47.9 | 39.4 | 15.5 |  46.5  | 29.4  | 35.8 |   35.8   |
| $\alpha$  = 0.6 | 47.8 | 39.8 | 16.5 |  45.6  | 29.1  | 35.1 |   35.7   |
| $\alpha$  = 0.8 | 47.6 | 36.4 | 16.5 |  43.6  | 26.7  | 35.4 |   34.4   |


α is the uncertainty factor used to balance the mean and standard deviation of the score from token-R. As α increases, Sentece-R tends to focus more on the uncertainty of LLMs, neglecting the average score which fundamentally reflects the sample quality. Therefore, α = 0.2 is an appropriate value to effectively balance sample quality and model uncertainty. We will add the results and discussion in the next version.

---

### Decision · Program_Chairs · 2024-09-25

**Decision:**

Accept (poster)

**Comment:**

This paper introduces a novel approach for instruction tuning of LLMs that leverages the model's intrinsic uncertainty to select high-quality training data without relying on external resources. The reviewers generally agree that the approach is innovative and potentially impactful. The strong empirical results across various benchmarks also support its effectiveness. The reviewers are largely positive on this submission. The authors also have addressed many concerns in their rebuttal, and provided additional experiments on diverse datasets and clarified the method's adaptability to different numbers of base models.